# Exploring the Optical Properties of Leaf Photosynthetic and Photo-Protective Pigments In Vivo Based on the Separation of Spectral Overlapping

**Yao Zhang** [1,2,3], **Chengjie Wang** [1], **Jingfeng Huang** [2,3], **Fumin Wang** [2,3], **Ran Huang** [1], **Hongze Lin** [1], **Fengnong Chen** [1] and **Kaihua Wu** [1,*]

1. College of Artificial Intelligence, Hangzhou Dianzi University, Hangzhou 310018, China; zhangyao@hdu.edu.cn (Y.Z.); wangchengjie@hdu.edu.cn (C.W.); ran_huang@hdu.edu.cn (R.H.); linhongze@hdu.edu.cn (H.L.); fnchen@hdu.edu.cn (F.C.)
2. Institute of Agricultural Remote Sensing and Information Application, Zijingang Campus, Zhejiang University, Hangzhou 310058, China; hjf@zju.edu.cn (J.H.); wfm@zju.edu.cn (F.W.)
3. Key Laboratory of Agricultural Remote Sensing and Information System, Hangzhou 310058, China
* Correspondence: wukaihau@hdu.edu.cn; Tel.: +86-571-87713528; Fax: +86-571-87713528

**Abstract:** The in vivo features of the absorption of leaf photosynthetic and photo-protective pigments are closely linked to the leaf spectrum in the 400–800 nm regions. However, this information is difficult to obtain because the overlapping leaf pigments can mask the contribution of individual pigments to the leaf spectrum. Here, to limit the masking phenomenon between these pigments, the separation technology for leaf spectral overlapping was employed in the PROSPECT model with the ZJU dataset. The main results of this study include the following aspects: (1) the absorption coefficients of separated chlorophyll a and b, carotenoids and anthocyanins in the leaf in vivo display the physical principles of forming an absorption spectrum similar to those in an organic solution; (2) the differences in the position of each absorption peak of pigments between the leaf in vivo and in an organic solution can be described by a spectral displacement parameter; and (3) the overlapping characteristics between the separated pigments in the leaf in vivo are clearly drawn by a range of absorption feature (RAF) parameter. Moreover, the absorption coefficients of the separated pigments were successfully applied in leaf spectral modeling and pigment retrieval. The results show that the separated multiple pigment absorption coefficients from the leaf spectrum in vivo are effective and provide a framework for future refinements in describing leaf optical properties.

**Keywords:** leaf optical properties; photosynthetic and photo-protective pigments; spectral overlapping separation; multiple pigment absorption features in vivo leaf

## 1. Introduction

Leaf spectra in the 400–800 nm region contain information about multiple photosynthetic pigments, including chlorophyll a (Chla), chlorophyll b (Chlb) and carotenoids (Cars), and photo-protective pigments, such as anthocyanins (Ants) [1]. These pigments are closely linked to the physiological and ecological functions of vegetation [2,3]. In particular, leaf photosynthetic electron harvest, transport and absorption are performed in the Chla and Chlb molecules [4], leaf fluorescence emission in the Chla molecules [5], leaf thermal dissipation of the xanthophyll cycle in Cars [6], and the leaf quenching of excess light energy in the Ants [7]. The retrievals of leaf pigments with remote sensing data are closely linked to the optical properties of leaf pigments in vivo. Thus, improving the determination of optical properties of leaf pigments in leaves in vivo can be achieved by better measurement and

knowledge of the pigments present in plant leaves, which should then provide a better understanding of the growth status of plants.

As a result of the selective light absorption of leaf pigments in the visible spectra, the color changes for a single leaf follow the proportion that its pigments present, which provide a chance to retrieve the content of leaf pigments by employing leaf reflectance/transmittance [8]. Although the selective absorption nature of leaf pigment groups raises its own absorption coefficients in leaves in vivo, the overlapping characteristics of these pigment groups raise the problem that the information of these pigment groups cannot easily be separated in the leaf spectra [9,10]. Ustin et al. summarized leaf spectral indices developed as chlorophyll, carotenoid and anthocyanin indicators and found the following [11]: (1) Single wavelength indices are rarely employed for higher chlorophyll content; (2) Combinations of wavelength indices for these three leaf pigment groups are more than other typical wavelengths indices; and (3) The research on leaf chlorophyll spectral index are maximal, followed by the carotenoid indices, and the anthocyanin indices are minimal. This could be owing to two facts: (1) the content of leaf chlorophyll is larger than that of the other leaf pigments, and (2) there are overlapping characteristics between the absorption spectra of leaf pigment in leaves in vivo. Thus, the separation of overlapping characteristics between the leaf pigment groups is a key factor for the retrieval of the content of leaf pigments.

Although the absorption coefficients of pure plant leaf pigments have been explored in organic solutions directly or with multiple linear regression method [12,13], these optical properties are difficult to use directly to describe the characteristics of pigment absorption in leaves in vivo. This is a difference between the corresponding peak positions of these absorption spectra in an organic solution and a leaf in vivo, e.g., the absorption peak position of the absorption of chlorophyll in the red range differs by approximately 20 nm in a specific organic solution and a leaf in vivo [10,14,15]. However, leaf radiative transfer models (RT models) could determine the property of absorption of leaf pigments in vivo and the biophysical characteristics of a leaf by modeling the optical actions of reflection, transmission and absorption of incident electromagnetic radiation [11,16]. Thus, the RT models also provided an opportunity to accurately analyze remotely sensed signals by quantifying the response of the optical properties of leaf pigment (pigment absorption coefficients) in a leaf in vivo [10,11]. Notably, PROSPECT (leaf optical PROperties SPECTra) [17] has become a key model to monitor the presence of plant pigments by modeling leaf optical properties in the 400–800 nm region. This model can effectively determine the objective pigment absorption coefficients, compared with other broadleaf RT models, e.g., the LEAFMOD (Leaf Experimental Absorptivity Feasibility MODel) models published by Ganapol et al. [18] and Berdnik [19] and SLOP (Stochastic Model for Leaf Optical Properties) by Maier et al. [20].

Several PROSPECT versions for the retrieval of leaf different pigment absorption coefficients, including PROSPECT-1 [17], PROSPECT-2 [21], PROSPECT-3 [22], PROSPECT-4 and 5 [23] and PROSPECT-D [24], have been released since 1990. They correspond to the separation of leaf total chlorophyll (Chls), Cars and Ants with a minimum distance fitting of spectra to optimize the calibration process of these pigment absorption coefficients [11]. These separated pigment absorption coefficients were adapted to account for leaf fluorescence, reflectance and the fluorescence emission of transmittance spectra, and then retrieve the content of Chls and Cars [25]. Although these developments of PROSPECT can accurately describe the contribution of leaf biophysical characteristics, the spectral overlapping feature between different individual pigments was still not considered. This could explain why the Chla and Chlb specific absorption coefficients could not be successfully separated and the determined Cars specific absorption coefficient is not following with the physical principles of the formation of Car absorption spectra in the PROSPECT-5 [23] and PROSPECT-D [24] models. Recently, an improved algorithm for the determination of multiple pigment absorption coefficients in leaves, employing a modified Gauss–Lorentz function that could simulate the physical mechanism of absorption spectrum produced by chromophore of photosensitive substance [26,27], enables the program to limit the masking phenomenon of different individual pigments owing to overlapping features of pigments, this was developed in the newest version of this model (PROSPECT-MP) that can determine the absorption

coefficients of Chla, Chlb and Cars [28]. The monitoring of plant physiological and ecological status and pigment discrimination requires much finer and more knowledge of the optical properties of leaf pigments in vivo, i.e., the simultaneous retrievals of Chla, Chlb, Cars and Ants from remote sensing data [11,29,30]. Thus, the improved algorithm of separating leaf multiple pigment absorption coefficients and the availability of a dataset (ZJU) with information on Chla, Chlb, Cars and Ants provides an opportunity to explore the optical properties of the leaf multiple photosynthetic and photo-protective pigments in vivo using the PROSPECT model.

To date, the optical properties of these individual pigments in the leaf in vivo have not been simultaneously explored and this has limited our capacity to retrieve individual pigment content from remotely-sensed spectra. Therefore, in order to describe the optical properties of Chla, Chlb, Cars and Ants of in vivo leaf, the present study simultaneously separated the in vivo absorption coefficients of multiple specific photosynthetic and photo-protective pigments in a leaf PROSPECT model through employing a modified Gauss–Lorentz function and a dataset (ZJU) with information on Chla, Chlb, Cars and Ants, importantly avoiding the masking phenomenon in model parameter separation and quantitatively describing the in vivo leaf pigment optical properties.

## 2. Materials and Methods

### 2.1. Data

#### 2.1.1. ZJU Dataset

The ZJU dataset [31] was collected from September to December in 2015. The leaf samples contained a range of 12 species with different biochemical and biophysical characteristics and different leaf ages (young, mature, senescent and albino leaves) that encompassed evergreen and deciduous trees, shrubs, subshrubs and herbaceous broadleaf plants.

Leaf biochemical and biophysical characteristics that have a wide range in the ZJU dataset are suitable for the analysis of leaf pigment coefficients in vivo that can be applied for the retrieval of major leaf pigments. The biochemical and spectral characteristics of the ZJU dataset are shown in Table 1 and Figure 1.

**Table 1.** Leaf biochemical measurement for the ZJU dataset.

| Leaf Pigment | Maximum | Minimum | Average | Unit |
|:---:|:---:|:---:|:---:|:---:|
| Chla | 94.53 | 0.04 | 24.63 | $\mu g/cm^2$ |
| Chlb | 47.49 | 0.05 | 12.75 | $\mu g/cm^2$ |
| Ants | 47.22 | 0.01 | 4.12 | $\mu g/cm^2$ |
| Cars [A] | 44.55 | 0.24 | 16.09 | $\mu g/cm^2$ |
| Lu | 17.71 | 0.02 | 4.76 | $\mu g/cm^2$ |
| An | 1.83 | 0.00 | 0.37 | $\mu g/cm^2$ |
| Ze | 6.99 | 0.02 | 1.06 | $\mu g/cm^2$ |
| Vi | 4.10 | 0.00 | 0.95 | $\mu g/cm^2$ |
| Ne | 7.43 | 0.00 | 1.85 | $\mu g/cm^2$ |
| β-car | 15.33 | 0.02 | 4.10 | $\mu g/cm^2$ |
| Water concentration | 73.83 | 11.61 | 52.34 | % |

Note that the superscript A expresses the Cars content as the sum of Lu, An, Ze, Vi, Ne and β-Car content in the corresponding leaf samples.

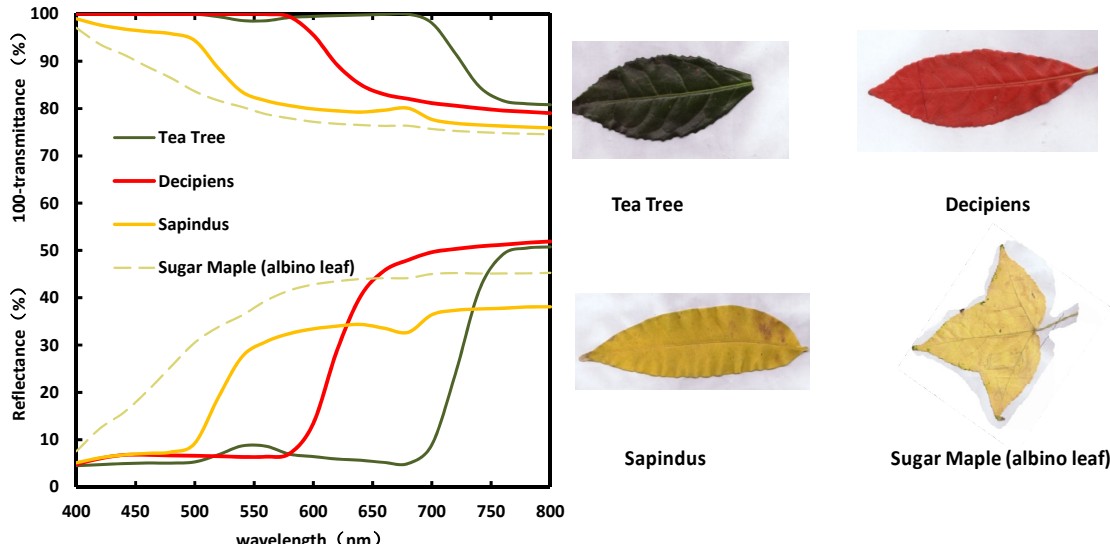

**Figure 1.** Reflectance and transmittance characteristics of the fresh leaf with the typical pigments from the ZJU dataset. Tee tree leaf pigment content (Cha: 78.34 μg/cm$^2$; Chb: 35.61 μg/cm$^2$; Cars: 12.22 μg/cm$^2$; Ants: 1.81 μg/cm$^2$), Decipiens (Cha: 0.6 μg/cm$^2$; Chb: 0.55 μg/cm$^2$; Cars: 0.05 μg/cm$^2$; Ants: 47.21 μg/cm$^2$), Sapindus (Cha: 1.15 μg/cm$^2$; Chb: 1.24 μg/cm$^2$; Cars: 0.92 μg/cm$^2$; Ants: 0.871 μg/cm$^2$); Sugar Maple (Cha: 0.11 μg/cm$^2$; Chb: 0.084 μg/cm$^2$; Cars: 0.02 μg/cm$^2$; Ants: 0.02 μg/cm$^2$).

### 2.1.2. Spectral Characteristics of the Absorption Spectra of Pure Pigments in Leaves

The absorption peak number and positions of the absorption spectra of pure leaf pigments (Chla, Chlb, β-Car (β-Carotene), Vi (Violaxanthin), An(Antheraxanthin), Ze(Zeinoxanthin), Ne(Neoxanthin), Lu(Lutein) and Ants) in organic solution are required for the expression of specific absorption coefficients function of the leaf pigments. A Shimadzu UV-VIS detector (350–800 nm) in a high performance liquid chromatography (HPLC) system was used to measure the absorption spectra of these pure pigments [32] in the mixed organic solution (see Figure 2). These pure pigments and the organic solutions were chromatography grade and were purchased from Sigma-Aldrich Co. (St. Louis, MO, USA). The purity of chlorophyll a is ≥90%, chlorophyll b is 95.4%, β-Car is 98.6%, Vi is 99.4%, An is 95.3%, Ze is 97.4%, Ne is 96.1%, Lu is 98.6% and Anths (Procyanidin) ≥90.0%. These spectral characteristics could be used for the separation algorithm of pigment absorption features with band overlapping in the leaf in vivo.

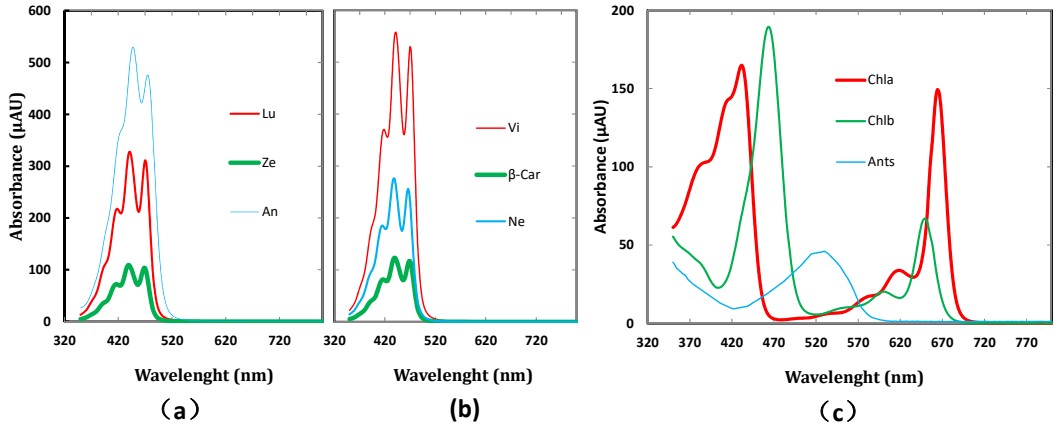

**Figure 2.** The absorption spectra of pure pigments in acetonitrile/methanol/dichloromethane (60:20:20 *v/v/v*, pH = 5) with a few 37% HCl solution. The content of Lu, An, Ze in (**a**) and Ne, Vi, β-Car in (**b**) were both 0.2 mg/mL and Chl*a*, Chl*b* in (**c**) were 0.01 mg/mL and Ants in (**c**) were 0.05 mg/mL.

## 2.2. Methods

### 2.2.1. Calibration of the Leaf Absorption Coefficient for PROSPECT-MP+

PROSPECT-MP is an extended version of PROSPECT in 400–800 nm by incorporating leaf multiple individual photosynthetic pigments (including Chla, Chlb and Cars) and the improved algorithm of separating leaf multiple photosynthetic pigment absorption coefficients [28]. In the improved algorithm, the absorption coefficients of Chla, Chlb and Cars with band overlapping features were successfully separated in the leaf in vivo, employing a modified Gauss–Lorentz function to limit the masking phenomenon from overlapping features. Similarly, there was an obvious band that overlapped between the absorption features of the Ants, Chlb, Cars and Chla [10,11]. To simultaneously retrieve the absorption coefficients of Chla, Chlb, Cars and Ants from the leaf spectra in vivo, we employed the improved algorithm to separate the absorption coefficients of multiple photosynthetic pigments and also simultaneously considered photo-protective pigment (Ants) in the in vivo leaves to extend PROSPECT-MP (PMP) to a new version: PROSPECT-MP+ (PMP+).

To simultaneously retrieve the absorption coefficients of Chla, Chlb, Cars and Ants, an extension of the leaf absorption coefficient for these pigments was necessary in the PROSPECT model. Based on the results from Feret et al. [23] and Zhang et al. [28], the leaf absorption coefficient ($k(\lambda)$) that incorporates Chla, Chlb, Cars and Ants is described in the following equation:

$$k(\lambda) = \frac{K_{Chla}(\lambda)C_{Chla} + K_{Chlb}(\lambda)C_{Chlb} + K_{Cars}(\lambda)C_{Cars} + K_{Ants}(\lambda)C_{Ants}}{N} + K_0(\lambda) \tag{1}$$

where $K_{Chla}(\lambda)$, $K_{Chlb}(\lambda)$, $K_{Cars}(\lambda)$ and $K_{Ants}(\lambda)$ indicate specific absorption coefficients for Chla, Chlb, Cars and Ants, respectively; $C_{Chla}$, $C_{Chlb}$, $C_{Cars}$ and $C_{Ants}$ indicate the content of Chla, Chlb, Cars and Ants in the corresponding fresh leaf, respectively; $K_0(\lambda)$ indicates the baseline absorption coefficient for the absorption characteristics of the non-pigment photosensitive material in the leaf in vivo, and N indicates the leaf structure index.

Since the pigment absorption feature originates from the sum of absorption feature of each peak, the pigment-specific absorption coefficients derive from the sum of all the absorption peaks in the corresponding individual pigments:

$$K_i(\lambda) = \sum_{j=1}^{j} K_{i,j}(\lambda) \tag{2}$$

where *i* is the determinable pigment type (Chla, Chlb, Cars and Ants), and *j* is the peak number within the pigment-specific absorption coefficient;

Based on the improved algorithm of the separation of absorption coefficients of multiple leaf pigments in PROSPECT-MP (Zhang et al., 2017) that could limit the overlap between the characteristics of leaf pigment absorption in the leaf spectrum in vivo, each absorption peak of these pigment absorption coefficients in Equation (3) was uniformly characterized by a modified Gauss–Lorentz (G–L) function:

$$K_{i,j}(\lambda) = K_{i,j,v} \cdot K_{i,j,h} \cdot e^{-4ln2 \cdot \left(\frac{A_{i,j,p} + K_{i,j,\Delta\lambda} - \lambda}{K_{i,j,w}}\right)^2} + \left(1 - K_{i,j,v}\right)\frac{K_{i,j,h}}{1 + 4(A_{i,j,p} + K_{i,j,\Delta\lambda} - \lambda)^2 K_{i,j,w}^{-2}} \tag{3}$$

where $K_{i,j}(\lambda)$ represents the *j*th peak function within the absorption coefficient for the *i*th pigment type; $K_{i,j,v}$, $K_{i,j,h}$ and $K_{i,j,w}$ are the Gauss ratio, peak height and full width at half maximum (FWHM) of the *j*th absorption peak for the *i*th pigment type in the leaf in vivo, respectively; $A_{i,j,p}$ is the peak position of the jth absorption peak for the ith pigment type in organic solution, and $K_{i,j,\Delta\lambda}$ is the spectral

displacement of the *j*th absorption peak for the *i*th pigment type in vivo leaf. The factors *i* and $A_{i,j,p}$ are shown in the Table 2 modified from Zhang et al. [28].

**Table 2.** The number and position of absorption peak for pure pigment in the 400–800 nm region from a mixed organic solution.

| Absorption Peak No. | $A_{Chla,j,p}$ (nm) | $A_{Chlb,j,p}$ (nm) | $A_{Cars,j,p}$ (nm) | $A_{Ants,j,p}$ (nm) |
|---|---|---|---|---|
| j = 1 | 432 | 458 | 418 | 530 |
| j = 2 | 580 | 602 | 443 | - |
| j = 3 | 618 | 650 | 470 | - |
| j = 4 | 664 | - | - | - |

As the PROSPECT-MP version, we applied the characterized $k(\lambda)$ of Equation (1) into the PROSPECT model at 400–800 nm, incorporating the in vivo leaf absorption coefficients of multiple photosynthetic (Chla, Chlb and Cars) and photo-protective pigments.

2.2.2. Determination of the Absorption Coefficients of Pigments in the Leaf In Vivo

In this study, the absorption coefficients ($K_i(\lambda)$ ) of Chla, Chlb, Cars and Ants in the leaves in vivo were determined using PROSPECT-MP+ by minimizing the merit function with a least-squares optimization:

$$\chi(K_i(\lambda)) = \sum_{m=1}^{m} \sum_{\lambda=400}^{800} (R_{mea}(\lambda) - R_{mod}(\lambda))^2 + (T_{mea}(\lambda) - T_{mod}(\lambda))^2 \tag{4}$$

where *m* stands for leaf sample number of the selected data for the determination from the ZJU dataset (*m* = 31), and the left leaf samples were used for the application evaluation (in Section 3.2, *n* = 28), and $R_{mea}$ and $T_{mea}$, $R_{mod}$ and $T_{mod}$ are the measured reflectance and transmittance, modeled reflectance and transmittance of the selected leaf samples, respectively. In the determination, the input variables contain the measured spectrum data ($R_{mea}$ and $T_{mea}$ ), pigment data ($C_{Chla}$, $C_{Chlb}$, $C_{Cars}$ and $C_{Ants}$) and the *N* and $K_0$ determined, in which the determinations of *N* and $K_0$ variables were dependent on the methods of Feret et al. [23] and Zhang et al. [28].

## 3. Results and Discussion

### 3.1. Optical Properties of the Absorption Coefficients of Pigments Determined by the Leaf In Vivo

3.1.1. Accordance with Their Physical Principles of the Formation of Absorption Spectra

Based on the absorption spectrum-forming principle of molecules or atoms in the visible regions [27], when the transitions of these molecules or atoms are in a type of skip or discontinuousness mode, their energy absorption is discontinuous. Moreover, their absorption spectra display discontinuous absorption peaks, and each absorption peak holds a normal distribution feature depending on the position of the absorption peak [33].

In this study, the G–L function, a linked normal distribution function with the explicit physical significance of material absorption spectra, was utilized to characterize the pigment absorption coefficients [34]. The $K_{Chla}$ and $K_{Chlb}$ that were determined have two obvious absorption peaks with their own absorption peak position (the 1st absorption peaks of $K_{Chla}$ with an absorption peak position of 419 nm ($K_{Chla,1,419}$ ), $K_{Chla,4,679}$ and $K_{Chlb,1,468}$ , $K_{Chlb,3,661}$ , respectively), and $K_{Cars}$ and $K_{Ants}$ have one obvious absorption peak with the absorption peak positions at 482 and 544 nm, respectively. The shapes of these pigment absorption coefficients that were determined have similar undulating shapes consistent with the absorption spectra of the corresponding pure pigment in the organic solution (Figure 3).

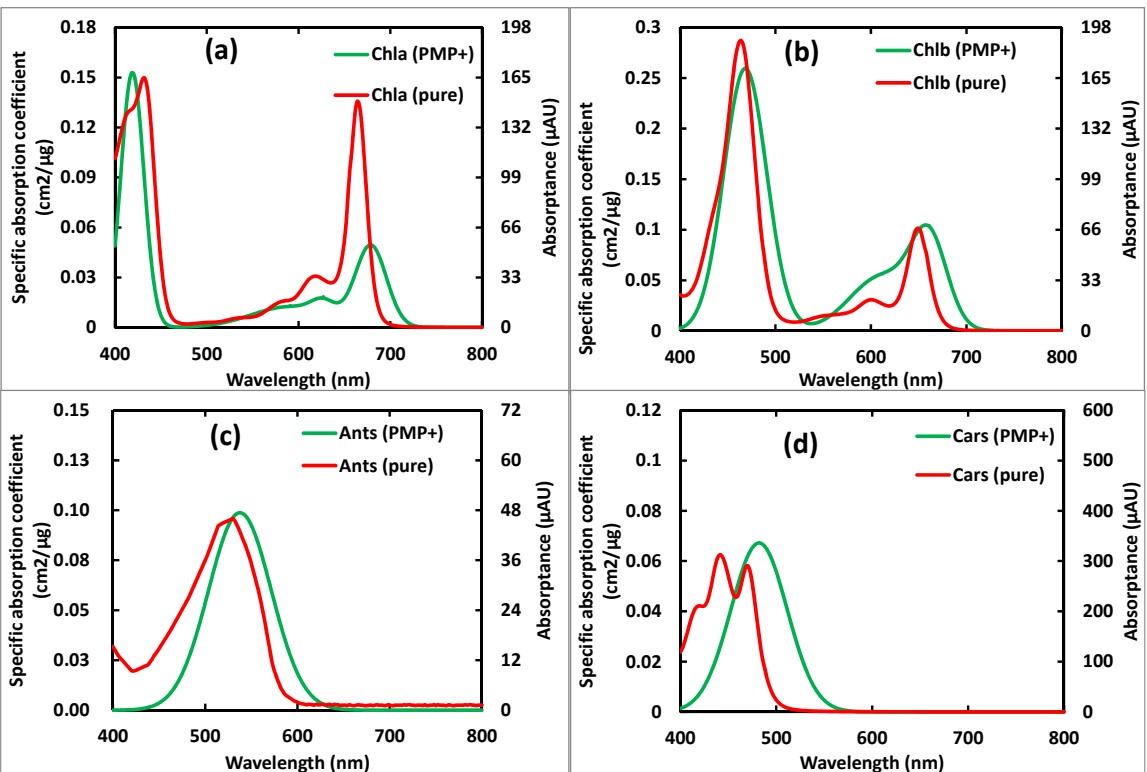

**Figure 3.** Optical property comparison of the determined pigment absorption coefficients in vivo leaf from PROSPECT-MP+ (PMP+) and the absorption spectra of the pure pigments in the organic solution from pure pigments: (**a**) Chla; (**b**) Chlb; (**c**); and(**d**) Ants.

### 3.1.2. Account of the Peak Position Variations Compared with the Specific Organic Solution

In Figure 3, the positions of the determined pigment absorption coefficients in the peak of absorption of the leaf in vivo using PROSPECT-MP+ (PMP+) emerge as different displacements related to the organic solution, i.e., the displacement of the $K_{Chla,1}$ position shifts to a shorter wavelength direction (designated a blue shift), and the $K_{Chla,4}$ position displaces to a longer wavelength (designated a red shift). In this study, a parameter, spectral displacement ($\Delta\lambda_{i,j}$), was used to account for variations in the peak position in the different polarity condition from the organic solution and an in vivo leaf using the spectral displacement parameter, i.e., the displacements of $K_{Chla,1}$ and $K_{Chla,4}$ were −13 nm (blue shift) and 15 nm (red shift), respectively; those of $K_{Chlb,1}$ and $K_{Chlb,3}$ were 4 and 11 nm, respectively, and those of $K_{Cars}$ and $K_{Ants}$ are 39 and 14 nm, respectively (see Table 3).

The average water concentration in the leaf in vivo is commonly more than 50% in the datasets from optical leaf experiments (LOPEX93, CALMIT, ANGERS and HAWAII) [23] and ZJU (see Table 1). In addition, the fresh leaves can be regarded as surrounded by a water medium. Marchetti et al. [35] reported the order of the polarity of different media: water > acetonitrile > methanol > dichloromethane. Thus, the environmental polarity of the leaf pigments determined in the leaf in vivo is more than that in the mixed organic solution (acetonitrile/methanol/dichloromethane; 60:20:20 *v/v/v*, pH = 5). Thus, there was a difference in polarity between the leaf in vivo and the organic solution.

Here, the different displacement involved in the blue and red shift of the individual pigment absorption peak could be caused owing to the effect of a difference in polarity on the electron transition mode of the chromophore between in the in vivo leaf and in organic solution. Thus, there was a blue shift or a red shift for the corresponding absorption peak position of the absorption spectra/coefficients of the leaf pigment between the leaf in vivo and the organic solution.

**Table 3.** Absorption peak characteristics determined from the in vivo pigment absorption coefficients. $K_{i,j,v}$, $K_{i,j,h}$ and $K_{i,j,w}$ are the Gauss ratio, peak height and full width at half maximum (FWHM) of the *j*th absorption peak for the *i*th pigment type in the leaf in vivo, respectively; $A_{i,j,p}$ is the peak position of the *j*th absorption peak for the *i*th pigment type; $K_{i,j,\Delta\lambda}$ is the spectral displacement of the *j*th absorption peak for the *i*th pigment type comparing with those in organic solution, and RAF is the range of absorption feature for each absorption peak of pigment absorption coefficients in the leaf in vivo.

| Specific Absorption Coefficient | Absorption Peak | $K_{i,j,v}$ | $K_{i,j,h}$ $(cm^2/\mu g)$ | $K_{i,j,w}$ (nm) | $K_{i,j,p}$ (nm) | $\Delta\lambda_{i,j}$ (nm) | RAF (nm) |
|---|---|---|---|---|---|---|---|
| $K_{Chla}$ | j = 1 | 0.80 | 0.153 | 51 | 419 | −13 | 400–434 |
| | j = 2 | 1.00 | 0.016 | 73 | 591 | 11 | - |
| | j = 3 | 0.78 | 0.008 | 82 | 627 | 9 | - |
| | j = 4 | 0.37 | 0.049 | 25 | 679 | 15 | 659–699 |
| $K_{Chlb}$ | j = 1 | 0.45 | 0.254 | 60 | 468 | 4 | 442–495 |
| | j = 2 | 0.75 | 0.017 | 42 | 612 | 9 | - |
| | j = 3 | 0.44 | 0.106 | 57 | 661 | 11 | 639–683 |
| $K_{Cars}$ | j = 1 | 0.5 | 0.067 | 56 | 482 | 39 | 447–517 |
| $K_{Ants}$ | j = 1 | 0.45 | 0.099 | 100 | 544 | 14 | 494–594 |

Note that Note that the symbol "-" stands for the negligible values in the RAFs because of the low absorbance values of these features; $K_{i,j,p} = A_{i,j,p} + K_{i,j,\Delta\lambda}$.

### 3.1.3. Quantification of the Main Absorption Features with an RAF Parameter

The G–L function can accurately characterize the material absorption spectra, since the parameters of this function hold the explicit physical significance of absorption spectra, in which the parameters $K_{i,j,w}$ (Full Width at Half Maximum [FWHM]) and $K_{i,j,p}$ can describe the main absorption feature of the absorption peak. Here, we employed the RAF (the range of absorption feature) parameter that drove $K_{i,j,w}$ and $K_{i,j,p}$ to describe the main features of the absorption coefficients of pigments that were determined [28]. In Table 3, the two obvious absorption peaks of $K_{Chla}$ are located in the regions of 400–434 nm and 659–699 nm. The obvious absorption peaks of $K_{Chlb}$ are at 442–495 nm and 639–683 nm. $K_{Cars}$ and $K_{Ants}$ are in 447–517 nm and 494–594 nm regions, respectively. Ustin et al. [11] reported that pigment spectral indices are structured with a central band (also designated an absorption band) and a reference band. The position of these absorption bands for the spectral indices of Chla, Chlb, Cars and Ants were meta-analyzed by Huang et al. [36]. These results were essentially consistent with the corresponding absorption range of the pigments observed in this study.

### 3.1.4. Exploration of Their Spectral Overlapping Feature

In Figure 4f, there are large overlapping features between the pigment absorption coefficients separated in the leaf in vivo based on PMP+. The largest overlap emerges between $K_{Chla,4}$ and $K_{Chlb,3}$. $K_{Cars}$ then overlaps with $K_{Chlb,1}$ and $K_{Ants}$, and finally, $K_{Ants}$ overlaps $K_{Chlb,1}$, $K_{Chlb,2}$, and there is a small degree of overlap between $K_{Ants}$ and all the absorption peaks of $K_{Chla}$. Alternatively, the spectral regions that overlap between these separated pigment absorption coefficients can also be described using the RAF parameters (see Table 3), i.e., the overlapping regions between $K_{Chla,4}$ and $K_{Chlb,3}$ are the 659–683 nm regions from the RAF of $K_{Chla,4}$ (659–699 nm) and the RAF of $K_{Chla,4}$ (639–683). These spectral overlapping features of the absorption coefficients of pigments that were determined and the maximum content of chlorophyll in the leaf in vivo provide the evidence to explain the pigment spectral index research that found that the leaf chlorophyll spectral indices are maximal, followed by the carotenoid indices, and the anthocyanin indices are minimal [11,29]. In this study, the absorption peak position and the RAF of the in vivo leaf pigment optical properties are effectively determined in a leaf radiative transfer model (see Table 3). This could provide a chance for research on the absorption of the emission of plant fluorescence and secondary fluorescence [37].

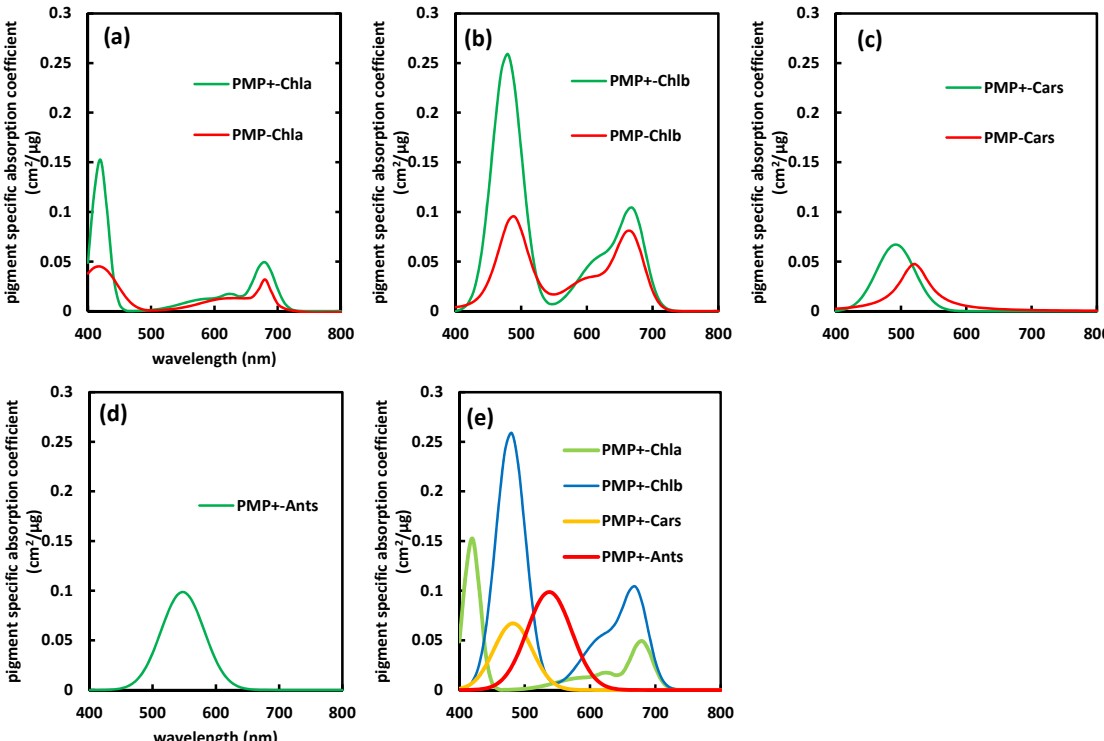

**Figure 4.** The spectral characteristics of the determined parameters in vivo for PROSPECT-MP (PMP) from Zhang et al. [28] and PROSPECT-MP+ (PMP+): (**a**) Chla specific absorption coefficient ($K_{Chla}$); (**b**) Chlb specific absorption coefficient ($K_{Chlb}$); (**c**) Cars specific absorption coefficient ($K_{Cars}$); (**d**) Ants specific absorption coefficient ($K_{Ants}$); and (**e**) the overlapping characteristics of the determined pigment absorption coefficients from PMP+.

In addition, compared with PMP (PROSPECT-MP), PMP+ (PROSPECT-MP+) can separate Chla, Chlb and Cars and can also produce more photo-protective pigment (Ants) absorption coefficient (in Figure 4). In addition, there are very similar positions of $K_{Chla,1}$, $K_{Chla,4}$ and the $K_{Chlb,3}$ between the PMP and PMP+ (see Figure 4, Table 3 and Zhang et al. [28]).

However, there are some differences between the pigment absorption coefficients ($K_{Chla}$, $K_{Chlb}$ and $K_{Cars}$) determined between PMP and PMP+. The main absorption regions of $K_{Chla}$, $K_{Chlb}$ and $K_{Cars}$ from PMP+ are higher than those from PMP. This is possibly owing to the differences that include the following: (1) PMP+ had a combined effect of the Ants, but this action is not accounted for PMP, in which the absorption feature of the Ants that overlapped other pigments was successfully separated in PMP+ and transferred to other pigments during the separation of pigment absorption coefficients in PMP, and (2) the measurement of the pigment content in ZJU utilized HPLC that can precisely determine the content of leaf pigments, and LOPEX93 [38] employed a spectrophotometric method that underestimated the content of Cars in the leaf [17]. For the absorption peak positions, there were visible differences in the first peak of $K_{Chlb}$ and the $K_{Cars}$ absorption peak between PMP and PMP+, which is the consideration of whether or not there were Ants in the two versions.

### 3.2. Evaluation of the Pigment Absorption Coefficients Determined in the In Vivo Leaf

#### 3.2.1. Analytical Evaluation of the Displacement of Peaks within the Absorption Coefficients

There are two sets of leaf reflections to evaluate the effectiveness of spectral displacement for the absorption peak positions in vivo compared with that of the organic solution. One set of leaves with a range of Ants content was extracted from the ZJU dataset. The second set of Chlb-deficient leaves was from a previous study [28]. In Figure 5a, the absorption by Ants dominated the 500–580 nm

regions, and the Absorption Peak Positions (APP) (545 nm) in the in vivo leaf were obtained by calculating the first and/or second derivatives of these reflection spectra. In Figure 5b, the absorption peak positions of Chla and Cars in an in vivo leaf are shown: $APP_{Chla,2} = 590$ nm, $APP_{Chla,3} = 628$ nm, $APP_{Chla,2} = 676$ nm, and $APP_{Cars} = 480$ nm, respectively.

The absorption peak positions of $K_{Ants}$, $K_{Cars}$, $K_{Chla,2}$, $K_{Chla,3}$ and $K_{Chla,2}$ from PMP+ were observed at 544, 482, 591, 627 and 679 nm (Table 3), respectively, which closely corresponded with the in vivo absorption peaks (Figure 5). This indicates that the process used to calibrate the pigment-specific absorption coefficients in PMP+ using ZJU data was effective.

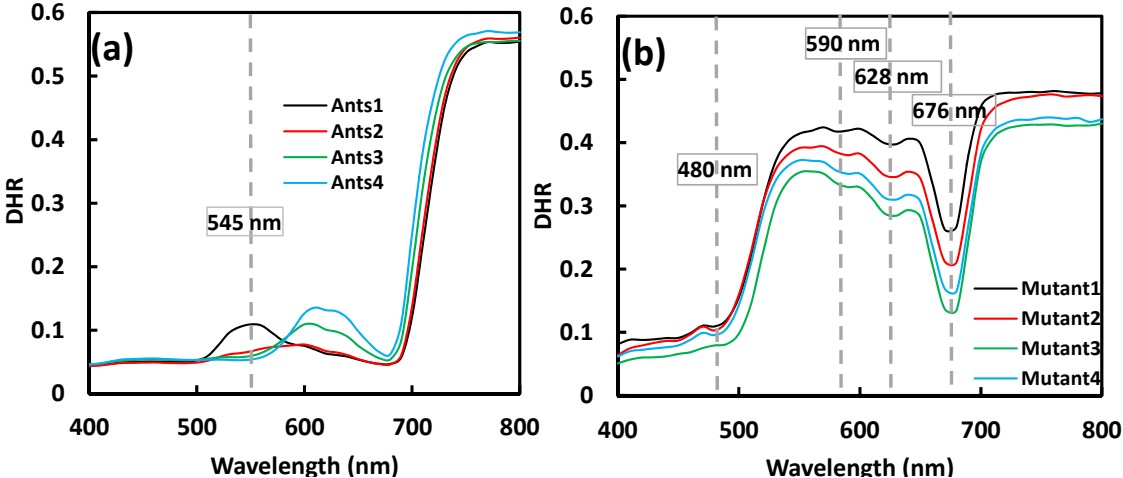

**Figure 5.** Leaf reflectance spectra reveal the in vivo pigment-specific absorption characteristics. Spectra are from (**a**) Red Robin leaves from ZJU with different Ants content (Ants1 = 2.379 μg/cm$^2$; Ants2 = 3.997 μg/cm$^2$; Ants3 = 6.117 μg/cm$^2$; and Ants4 = 11.036 μg/cm$^2$), and (**b**) is from the report [28]. DHR is leaf directional hemispherical reflectance.

### 3.2.2. Application Evaluation of Spectral Modeling and Pigment Retrieval

In this section, to evaluate the effectiveness of the determined pigment absorption coefficients, we applied these pigment absorption coefficients for the spectral modeling and pigment retrieval capabilities in PROSPECT-MP+ using the left leaves of the ZJU dataset (these leaf samples were not used for the determination of pigment absorption coefficients). Moreover, to describe the effect of Ants present on other pigment optical properties, we also compare the spectral modeling and pigment retrieval capabilities of pigment absorption coefficients from PROSPECT-MP (PMP) and PROSPECT-MP+ (PMP+). The metrics RMSE (Root Mean Square Error), BIAS (Bias), and SEC (Standard Error Corrected) were used for spectral modeling evaluation, and RMSE, BIAS, SEC and CV (Coefficient Variability) were for pigment retrieval evaluation [23].

Spectral Modeling

Figure 6 shows simulated and measured DHR and DHT spectra for leaves with the low, medium and high Ants content. The performance from PMP+ is particularly effective for the low Ants leaves, and it is encouraging for the medium Ants leaves. For the high Ants leaves there is some underestimation of DHR around 500–580 nm and overestimation at 650–700 nm, while the DHT simulation matches well with the measured spectrum. The spectral modeling performance from PMP is weaker than PMP+ across all three different Ants content. This can be attributed to the absence of an Ants absorption coefficient within the τ parameter in PROSPECT-MP [28].

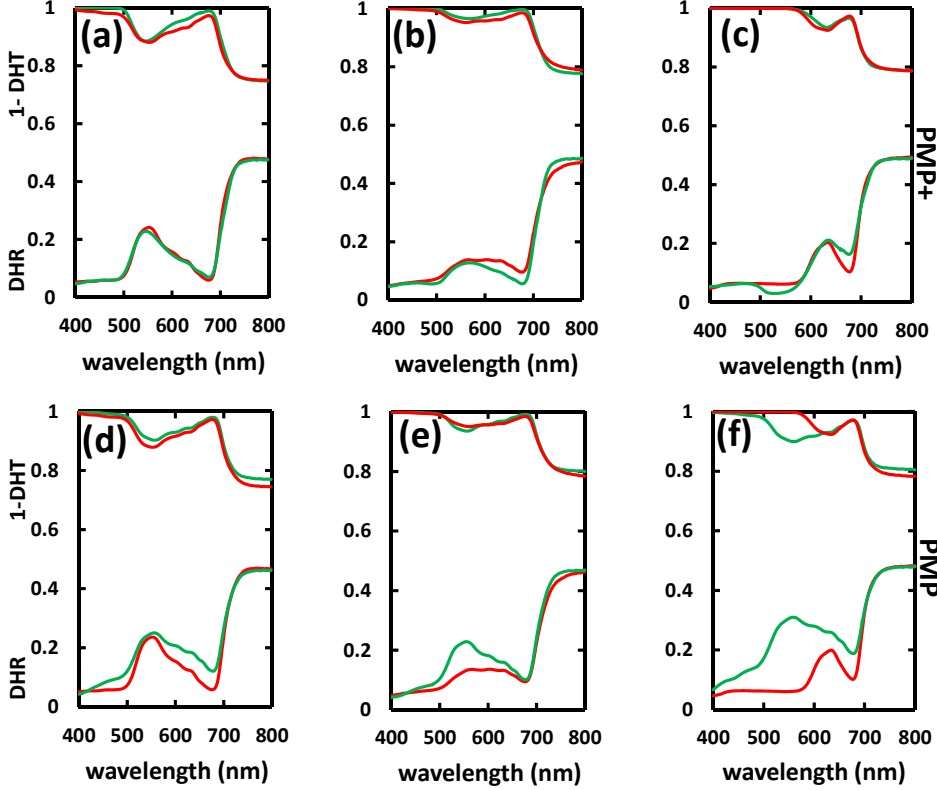

**Figure 6.** Comparison of measured (red) and simulated (blue) reflectance and transmittance spectra for the different Ants content leaves from PROSPECT-MP+ (PMP+) and PROSPECT-MP (PMP), in which (**a**,**d**) from the low, (**b**,**e**) from the medium, and (**c**,**f**) high from the high. DHR and DHT are leaf directional hemispherical reflectance and transmittance, respectively.

The global performance of PMP+ for the leaf DHR and DHT modelling is excellent, since RMSE and SEC are both less than 0.03, and BIAS is lower than ±0.01 (see Table 4). The PMP scores were lower than those in the PMP+ in every evaluation metric. The results indicate that PMP+ has a superior capability for leaf spectral modeling, but PMP is much less effective. These results confirm that PMP+ can successfully simulate the leaf spectra by incorporating the information on Ants, while PMP lacks this ability.

**Table 4.** Global performance evaluation of simulated leaf spectra from PROSPECT-MP+ (PMP+) and PROSPECT-MP (PMP) (leaf sample number *n* = 28). The metrics RMSE, BIAS and SEC are Root Mean Square Error, Bias and Standard Error Corrected for the errors between the measured and modeling spectra, respectively.

| Spectrum Type | Model Implementation | RMSE | BIAS | SEC |
|:---:|:---:|:---:|:---:|:---:|
| **DHR** | **PMP+** | 0.027 | 0.004 | 0.026 |
| | **PMP** | 0.046 | 0.027 | 0.036 |
| **DHT** | **PMP+** | 0.021 | −0.007 | 0.019 |
| | **PMP** | 0.026 | 0.007 | 0.025 |

In considering the local performances in spectral modeling, the largest errors generated by both PMP+ and PMP for the DHR and DHT simulations are located in the 500–600 nm region (Figure 7). With respect to PMP, this can be explained because, as alluded to above, PMP does not incorporate an Ants absorption coefficient, and the RAF of this pigment group is located in the 500–600 nm regions (Table 4). For PMP+, the larger errors are located at 540–580 nm and 710–800 nm. Although we have

considered the non-pigment and absorption of Ants in this implementation of PMP+, the modeling capability in these two spectral regions is not significantly improved compared with that of PMP [28]. It is possible that further improvements may require a more accurate determination of the average leaf refractive index using a complex refractive index [30].

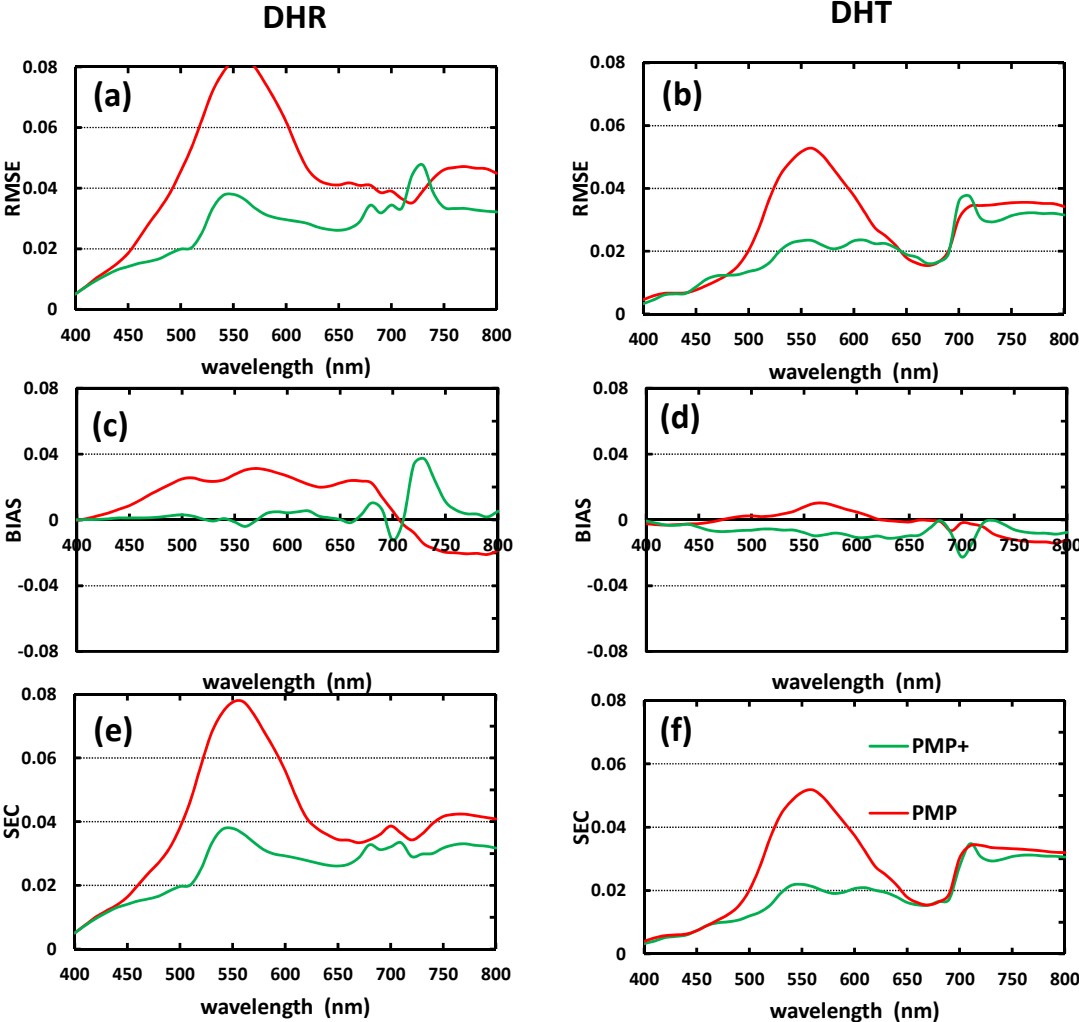

**Figure 7.** The valuation for simulated DHR (Directional Hemispherical Reflectance) and DHT (Directional Hemispherical Reflectance) spectra from PROSPECT-MP+ (PMP+) (black line, leaf sample number *n* = 28) and PSROPCT-MP (PMP) (grey line, *n* = 28): (**a**,**b**) for RMSE, (**c**,**d**) for BIAS, and (**e**,**f**) for SEC. The metrics RMSE, BIAS and SEC are Root Mean Square Error, Bias and Standard Error Corrected for the errors between the measured and modeling spectra, respectively.

Pigment Content Retrieval

In Figure 8, the capabilities of PROSPECT-MP+ (PMP+) and PROSPECT-MP (PMP) to retrieve pigments are illustrated. These results demonstrate that PMP+ can retrieve not only the content of Chla, Chlb and Cars from in vivo leaf DHR and DHT (as does PMP) but also simultaneously retrieve the sub-divisible photosynthetic pigments (Chla and Chlb) and photo-protective pigments (Ants). Moreover, the scatter points from Figure 8a–c are closer to the 1:1 line than those in Figure 8e–g, which demonstrates that PROSPECT-MP+ can improve the capability of the retrieval of content s of leaf Cars, Chlb and Chla compared with that of PMP. Ants is considered in PMP+, and the accuracy of the retrieval of Cars improved. Moreover, the absorption characteristics of Cars overlap those of Chla and Chlb. Thus, the improvement of accuracy of the retrieval of Cars helps the capability of retrieval of Chla and Chlb in PMP+.

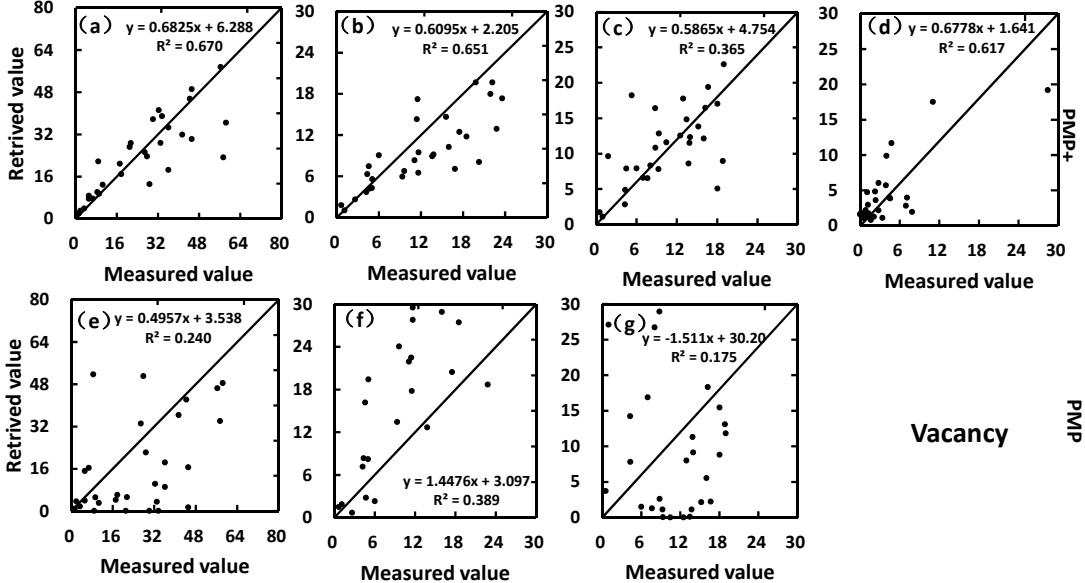

**Figure 8.** Comparison between measured and retrieved pigment content ($\mu g/cm^2$, (leaf sample number *n* = 28) from PROSPECT-MP+ (PMP+) and PROSPECT-MP (PMP). (**a**,**e**) are for Chla; (**b**,**f**) for Chlb; (**c**,**g**) for Cars; and (**e**) for Ants.

Compared with PMP, PMP+ can substantially improve the retrieval of Chls and Cars shown by the metrics of RMSE, SEC and CV (Table 5). PMP+ can accurately retrieve content s of Chla and Chlb from the leaf spectra in a manner similar to those previously reported [28]. Table 5 also shows that PMP+ is particularly effective for retrieving content s of Ants. It is also worth noting that PMP improved on the retrieval of content s of Chls and Cars compared with the reports of PMP in the LOPEX93 dataset, which may indicate that the measurement of photosynthetic pigments with HPLC (in the ZJU dataset) can improve the capabilities of PROSPECT-MP.

**Table 5.** The validation of pigment content retrievals from in vivo leaf spectra by PROSPECT-MP+ (PMP+) and PROSPECT-MP (PMP) (leaf sample number *m* = 28). The metrics RMSE, BIAS, SEC and CV are Root Mean Square Error, Bias, Standard Error Corrected and Coefficient Variability for the errors between the measured and retrieved content s of leaf pigment, respectively.

| Performance Types | PMP+ | | | | PMP | | |
|---|---|---|---|---|---|---|---|
| **Pigment Types** | **Chla** | **Chlb** | **Cars** | **Ants** | **Chla** | **Chlb** | **Cars** |
| **RMSE ($\mu g/cm^2$)** | 11.69 | 6.54 | 8.18 | 3.17 | 18.31 | 10.26 | 28.75 |
| **BIAS ($\mu g/cm^2$)** | −0.16 | −3.22 | 0.76 | 0.07 | −8.09 | 6.69 | 7.50 |
| **SEC ($\mu g/cm^2$)** | 11.69 | 5.67 | 8.15 | 3.17 | 16.39 | 7.73 | 27.74 |
| **CV (%)** | 31.84 | 39.37 | 39.24 | 45.42 | 65.66 | 67.26 | 269.81 |

## 4. Conclusions

This paper produces a new set of multiple photosynthetic and photo-protective pigment absorption coefficients using the ZJU dataset in a leaf optical radiative transfer model. The pigment absorption coefficients determined in an in vivo leaf also possess three key features: (1) the separated absorption coefficients of chlorophyll a and b, carotenoids and anthocyanins in an in vivo leaf display the physical principles of absorption spectrum-forming like those in organic solution; (2) the differences in the position of each absorption peak of pigments between the in vivo leaf and an organic solution can be described by a spectral displacement parameter; and (3) the overlapping characteristics between the separated pigments in the in vivo leaf are clearly drawn by a range of absorption feature parameter.

To provide some context, some of the absorption peak positions of chlorophyll a, carotenoids and anthocyanins in the in vivo leaf were demonstrated to be effective from two sets of specific leaf reflection spectra. Moreover, the capabilities of leaf spectral modeling and inversion for PROSPECT-MP+ were compared with those of PROSPECT-MP. The results were encouraging based on the following conclusions: (1) PROSPECT-MP+ can improve the simulation of the in vivo leaf DHR and DHT spectra, particularly in leaves with anthocyanins present; (2) PROSPECT-MP+ can improve the accuracy of retrieval of carotenoids, considering the band overlapping features between the absorption spectra of the carotenoids and anthocyanins; (3) PROSPECT-MP+ also provides the capability to reliably retrieve individual chlorophyll a and chlorophyll b content like PROSPECT-MP; and (4) PROSPECT-MP+ can also provide a means to accurately retrieve photo-protective pigment content, such as anthocyanins, from fresh leaf spectra.

Our ongoing work is now focused on improving the description of optical properties of leaf pigment in the in vivo leaf, with explicit parameterizations of the effect of leaf polar environment (e.g., leaf pH values and water concentration) on the red shift or blue shift of the position of leaf pigment absorption peak. Thus, these future developments of leaf pigment optical properties should improve the robustness and transferability in the capabilities to retrieve multiple pigment content and the synthesis of leaf pigment optical properties with plant fluorescence will offer opportunities to improve the estimates of vegetation physiological and ecological functions.

**Author Contributions:** Y.Z., J.H. and K.W. conceptualized the idea of the study; R.H. annotate, scrub and maintain research data; C.W. analyzed the results; Y.Z. supported for the project; Y.Z. performed the experiments; Y.Z., F.C. and F.W. design the methodology; K.W. and Y.Z. managed the research activity; C.W. supported study materials and other analysis tools; H.L. developed the software. K.W. leaded the research activity planning. C.W. and R.H. produced the validation dataset; C.W. prepared for the published work; Y.Z. and C.W. wrote the manuscript; Y.Z. and K.W. contribute to the revision of the manuscript. All authors have read and agreed to the published version of the manuscript.

**Funding:** This research was funded by National Natural Science Foundation of China, grant number 41471277; Zhejiang Natural Science Foundation of China, grant number LY20D010004, LQ20F050006; Zhejiang public welfare program of agriculture technology, grant number LGN18F030002; Major Agricultural Technology Cooperative Popularization Project of Zhejiang Province of China, grant number 2020XTTGCY01-02.

**Conflicts of Interest:** The authors declare no conflict of interest.

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
