# Peer review of "Exploring the Optical Properties of Leaf Photosynthetic and Photo-Protective Pigments In Vivo Based on the Separation of Spectral Overlapping"

_remotesensing, doi:10.3390/rs12213615_

Round 1
Reviewer 1 Report
The manuscript captures an established, but still innovative study area, with potential applications in agriculture, environmental recording, and forestry. The approach proposed is interesting, but the authors should reread the work on multiple linear regression applied for pigment separation (c.f. Pflanz and Zude, 2008 in Applied Optics) other than in remote sensing. Language needs to be revised throughout the manuscript - frequently the sentences have no or misleading meaning (not further marked in the text despite the two examples in lines 53, 59-61).
Specific remarks:
I would appreciate a space between text and brackets always.
Replace "Leaf pigment concentration" by "leaf pigment content", since a concentration can only be found in solution.
Line 53: please revise the langue: an effect cannot be "removed"
Lines 59-61: The language need revision here and throughout the manuscript and particularly in the first part of introduction before the Prospect is explained..
Place objectives at the end of the introduction, instead of stating what was done.
Table 1: a concentrtion cannot be given per cm².
Does "pure" pigment refer to standards?
Figure 2: space between value and unit. Why was HCl added? Was it added to anthocyanin solutions only?
Remove text between 2.2 Methods and 2.2.1.
Same for text between 3.1 and 3.1.1
Please explain unit cm²/µg.
Remove text between 3.2 and 3.2.1.
Figure 5: DHR needs to be explained in the figure caption below the figure.
Same for DHR and DHT in figure 6.
Table 4: Explain abreviations.
Find a better axis label than inversed in figure 8.
Author Response
Dear Reviewer:
Thank you for your kindly making some valuable comments and suggestions to our paper entitled “Exploring the optical properties of leaf photosynthetic and photo-protective pigments in vivo based on the separation of spectral overlapping” (remotesensing-947376). Uploaded please find our revised paper and this response letter. We have carefully studied comments from all reviewers and editors, responded to these comments point-by-point, and revised the manuscript accordingly. An independent response letter that includes a detailed point-by-point response will be uploaded in system. In the revised manuscript, all changes except for language revision made to the text are in red so that they are easily identified. We look forward to a favorable response from you soon.
Sincerely yours,
Kaihua Wu

Reviewer 2 Report
I found the manuscript difficult to read. There are many mistakes and ambiguity in the text, therefore it is difficult to understand what the authors want to communicate. For example, sentences in the following lines require some revision:
31-32; 48-49;50-53; 56; 57; 58; 61-62;65-68;72;74;87;89-93;93-95;99-111;117;133-134;137-138;173;181;195-198;245;261-264;266;268-271;320;327;396.
After revising the language, the manuscript should be reviewed one more time.
Below there are a few drawbacks which I could identify in the current version of the manuscript:
L330: please explain what are DHR and DHT;
L342-343: please explain what are RMSE, SEC and BIAS; please check why you write DHT and DHT.
Please introduce the abbreviations PMP and PMP+;
Please explain how do you minimize the expression (4), is it linear or non-linear by Ki?
Please explain better what is the difference between PMP and PMP+, what exactly did you change?
Author Response

(The authors gave the same response as above.)

Reviewer 3 Report
Dear authors,
the experiment is well designed and the paper well written. Thus, I suggest just few recommendations before publications, following listed:
- Which is the novelty of your work compared to previous research works? Please, enhance it in the introduction section;
- Introduce a detailed description of RS methods commonly applied to detect biochemical features of leaf? Which are the pros and cons of each of them? Why did you apply the PROSPECT-MP method?
- Compare your results with that ones obtained from previous work? Are they comparable?
- Edit the manuscript according to journal style
- Check the equations
- Improve Figure 6 using colours to distinguish the various lines
- Improve Figure using colours to distinguish the various lines
Author Response

(The authors gave the same response as above.)

Round 2
Reviewer 2 Report
The authors have corrected the text according to my suggestions and the suggestions by the other Reviewer. The manuscript can be accepted for publication.
Author Response
Dear Reviewer:
Thank you for kindly providing a comment.
Sincerely yours,
Kaihua Wu